# Effect of the Addition of Selected Herbal Extracts on the Quality Characteristics of Flavored Cream and Butter

**DOI:** 10.3390/foods12030471

**Published:** 2023-01-19

**Authors:** Małgorzata Ziarno, Mariola Kozłowska, Katarzyna Ratusz, Rozeta Hasalliu

**Affiliations:** 1Division of Milk Technology, Department of Food Technology and Assessment, Institute of Food Science, Warsaw University of Life Sciences-SGGW (WULS-SGGW), 02-787 Warsaw, Poland; 2Department of Chemistry, Institute of Food Science, Warsaw University of Life Sciences-SGGW (WULS-SGGW), 02-787 Warsaw, Poland; 3Division of Fats and Food Concentrates Technology, Department of Food Technology and Assessment, Institute of Food Science, Warsaw University of Life Sciences-SGGW (WULS-SGGW), 02-787 Warsaw, Poland; 4Faculty of Biotechnology and Food, Agricultural University of Tirana, 1029 Tirana, Albania

**Keywords:** savory, basil, oregano, rosemary, thyme

## Abstract

Herbs have been used for centuries in order to enrich food as preservatives, flavorings, and medicinal agents. The aim of this work was to study the effect of the addition of selected herbal extracts (dried leaves of *Thymus vulgaris* L., *Origanum vulgare* L., *Satureja hortensis* L., *Rosmarinus officinalis* L., and *Ocimum basilicum* L.) on selected parameters of fermented flavored cream (counts of starter culture bacteria and pH value) and the resulting flavored butter (water content, water distribution, butter plasma pH, butter fat acidity, and oxidative stability), preceded by a study of the activity of the herbal extracts against starter lactic acid bacteria determined using the well diffusion method. The extracts did not inhibit the starter lactic acid bacteria at a fixed level. The presence of the herbal extracts contributed to a shorter fermentation course and influenced the counts of starter culture bacteria during fermentation and refrigerated storage (at 5 °C) for 21 days. The extract additives did not affect the water content or the degree of its dispersion, the butter plasma pH, or the butter fat acidity. The positive effect of the rosemary and thyme extract addition was only noted when analyzing the oxidative stability of the milk fat of the butter.

## 1. Introduction

In recent decades, consumer requirements for food production have changed significantly. Today, food is designed not only to satisfy hunger and provide people with essential nutrients, but also to prevent diet-related diseases and improve people’s physical and mental well-being. Meanwhile, herbs have been used for centuries to enrich food as preservatives, flavorings, and medicinal agents [1,2]. Today, herbs and spices are used as food additives around the world, not only to improve the organoleptic properties of food, but also to extend shelf life. Due to their antimicrobial and antioxidant properties, they are useful in the dairy industry as a flavor fixative [3]. It has become apparent that lipids can be effectively protected from enzymatic hydrolysis, oxidation, and other adverse transformations by a variety of natural spices, sometimes more effectively than by synthetic antioxidants. Therefore, the addition of herbal spices to butter may now become a popular way to introduce safe antioxidant substances into this fat. Previous studies on the isolation of oregano (*Origanum vulgare* L.) essential oils from various regions of the world have focused on the chemical composition, although the antioxidant and antimicrobial properties have also been studied [4,5,6,7,8,9,10,11,12]. Herbal extracts have powerful antioxidant and antimicrobial effects, mainly due to the amount and quality of the phenolic compounds they contain [5,7,8,10]. Savory (*Satureja hortensis* L.) herb has significant antimicrobial activity, which is linked to the presence in the herb of an oil rich in carvacrol, a phenolic component that exhibits strong antimicrobial activity [13,14]. Thyme herb (*Thymi herba*) exhibits a broad spectrum of activity against the growth of many strains of Gram-positive and Gram-negative bacteria [15,16]. Common basil (*Ocimum basilicum* L.) extract could quench DPPH+ radicals and result in a high degree of inhibition of linoleic acid oxidation [17,18]. In addition, rosemary (*Rosmarinus officinalis* L.) is a plant rich in compounds that exhibit biological activity [3,19].

Milk and dairy products are unique carriers successfully used to deliver the bioactive ingredients that benefit human health. Furthermore, the addition of herbs and spices, or their extracts, to various dairy products enables these products to function as carriers of nutraceuticals [1,2].

The aim of this work was to study the effect of the addition of selected herbal extracts on selected quality characteristics of fermented flavored cream and the resulting flavored butter. We aimed to study the effect of the addition of the whole extract of the selected herbs, but not the specific compounds isolated from them.

## 2. Materials and Methods

### 2.1. Materials

The material for the study included extracts of commercially available dried leaves of popular herbs: thyme (*Thymus vulgaris* L.), oregano (*Origanum vulgare* L.), savory (*Satureja hortensis* L.), rosemary (*Rosmarinus officinalis* L.), and basil (*Ocimum basilicum* L.). The extracts were prepared by extracting dried leaves according to the protocol previously described in Kozłowska et al. [20], with some modifications. The preparation of the extract from the different herbs was the same. In general, per 10 g portion of dried herb leaves, an amount of 250 mL of 70% aqueous ethanol solution was used (it is noteworthy that ethanol is an environmentally friendly and safe solvent for human and is well suited for the extraction of phenolic compounds). This mixture was heated for 10 h in a water bath at 45 °C. Next, the solid plant residues were drained on Whatman No. 1 filter paper, while the ethanol was evaporated under vacuum in a Rotavapor R-200 rotary evaporator (Büchi Labortechnik, Flavil, Switzerland). The procedure for herbal extracts manufacture was carried out until sufficient quantities of lyophilized extracts had been collected to carry out the experiments described below. The resulting herbal extracts were lyophilized (Alpha 1-4 LSCplus, Osterode am Harz, Germany) and then stored tightly closed at −21 °C in a darkroom until used in experiments.

The raw material in the experiments was fresh Łowicka UHT cream with a fat content of 36% (OSM Łowicz, Łowicz, Poland). It was fermented using two industrial lyophilized dairy starter cultures: YC-X16 (from Chr. Hansen, containing *Streptococcus thermophilus* and *Lactobacillus delbrueckii* subsp. *bulgaricus*) and YO-MIX 207 (from Danisco, containing *Streptococcus thermophilus*, *Lactobacillus delbrueckii* subsp. *bulgaricus*, *Lactobacillus acidophilus*, and *Bifidobacterium lactis*). All chemicals used were of chemical purity and were purchased from BioMaxima (Lublin, Poland), Fluka (Merck KGaA, Darmstadt, Germany), or Merck (Merck KGaA, Darmstadt, Germany).

### 2.2. Agar Well Diffusion Method of Evaluating Antimicrobial Activity of Herbal Extracts

The agar well diffusion method is widely used to evaluate the antimicrobial activity of plant extracts and microbial extracts [4,6,8,9,10,11,12]. The surface of the medium plate was inoculated by spreading 1 mL of inoculum of starter culture bacteria (hydrated from lyophilized starter culture and revived after overnight incubation in nutrient broth, whereby the concentration of starter culture microorganisms was approximately 7–8 log cfu/mL) over the entire surface of the medium. Next, holes with a diameter of 6 mm were cut aseptically with a sterile cork borer, and 20 µL of herbal extract solution at the desired concentration was introduced into the well. The herbal extract solution was prepared in two versions: in sterile DMSO (dimethyl sulfoxide) and in a sterile 68% aqueous ethanol solution. Solutions were prepared by weighing 0.01 mg of each lyophilized herbal extract into 1 mL of each solvent. In order to perform the experiment, previously prepared Petri dishes with appropriate medium were prepared: M17 agar (BioMaxima, for *Streptococcus thermophilus*); De Man, Rogosa, and Sharpe agar (MRS agar, BioMaxima, for *Lactobacillus* spp.); MRS-CC agar (BioMaxima, with clindamycin at 0.5 mL/L and ciprofloxacin at 5.0 mL/L, for *Lactobacillus acidophilus*); and Bifidus Selective Agar (BSM agar, Fluka, with BSM supplement, for *Bifidobacterium lactis*) [21]. The agar Petri dishes were then incubated at 37 °C for 72 h under appropriate aerobic conditions, depending on the microorganism (Petri dishes with MRS agar, MRS-CC agar, and BSM agar under anaerobic conditions in anaerobic culture containers; Petri dishes with M17 agar under aerobic conditions). The antimicrobial substances present in the tested extracts diffused in the agar media and inhibited the growth of the tested microorganisms, and after incubation in the Petri dishes, the zones of inhibition (mm) of bacterial growth around the excised wells were measured with a digital caliper. The analysis was conducted in duplicate.

### 2.3. Fermentation and Testing of Fermented Cream

#### 2.3.1. Fermentation of Cream and Fermentation Curves of Cream

The cream was fermented in portions of 150 g in sterile glass jars. Before fermentation, 1.5 mg tested lyophilized herbal extracts and 1 mL sterile suspension of starter cultures (previously hydrated for 20 min by dissolving 5 g lyophilizate in 10 mL sterile Ringer’s solution) were added to each cream portion. A control sample was also prepared for each variant of the starter culture, which did not contain the addition of herbal extract. The cream samples prepared in this way were kept in an incubator at 45 °C for 6 h to ferment the cream. After the fermented cream manufacture, the samples were left in the refrigerator at 5 °C for 21 days for further analysis. Microbiological and physicochemical analysis of the fermented cream samples was conducted at 0, 7, 14, and 21 days of sample storage. The experiment was conducted in duplicate.

During the fermentation of the cream, samples were taken every hour to measure the pH value. The pH value of the cream samples was measured at 25 °C using a CPO-505 pH meter (Elmetron, Zabrze, Poland) with a conventional electrode probe [21]. The instrument was calibrated using buffer solutions of pH 4.00 and 7.00 at 25 °C. The analysis was conducted in duplicate.

#### 2.3.2. Counts of Starter Culture Bacteria and Contaminating Microflora during the Fermentation of the Cream and During Storage

Every two hours of the fermentation of the cream and every seven days of the refrigerated storage of the fermented cream, samples were taken to determine the counts of starter culture bacteria. The bacterial counts of the starter cultures were determined in the: M17 agar (BioMaxima, for *Streptococcus thermophilus*); De Man, Rogosa and Sharpe agar (MRS agar, BioMaxima, for *Lactobacillus* spp.); MRS-CC agar (BioMaxima, with clindamycin at 0.5 mL/L and ciprofloxacin at 5.0 mL/L, for *Lactobacillus acidophilus*); and BSM agar (Fluka, with BSM Supplement, for *Bifidobacterium lactis*) using the droplet method [21]. The Petri dishes were incubated at 37 °C for 72 h under appropriate aerobic conditions, depending on the microorganism (Petri dishes with MRS agar, MRS-CC agar, and BSM agar under anaerobic conditions in anaerobic culture containers; Petri dishes with M17 agar under aerobic conditions). After the incubation was completed, the grown colonies were counted. The result was given in colony-forming units of 1 mL (cfu/mL) and then expressed as the logarithm of the total number of bacteria cells. The analysis was conducted in duplicate.

At the same time as measuring the counts of starter culture bacteria, the presence of contaminating microflora was measured. The total number of *Enterobacteriaceae* was determined in VRBG medium (with crystal violet, neutral red, bile salts, and glucose, Merck) overlaid on the same medium and incubated at 37 °C for 24 h. The total numbers of molds and yeasts were determined in YGC medium (yeast extract glucose chloramphenicol agar FIL-IDF, Merck), and the plates were incubated at 25 °C for 5 days. The result was given in colony-forming units of 1 mL (cfu/mL). The analysis was conducted in duplicate.

### 2.4. Butter Manufacture and Testing

#### 2.4.1. Butter Manufacture

The freshly fermented cream was subjected to the buttering process in a laboratory butter churner machine type 83 (Zelmer, Warsaw, Poland). The fermented cream was brought to a temperature that would allow the buttering process to proceed (at 8 °C), and the buttering process was conducted in a butter churner machine until butter grains formed (approximately 45–60 min). The buttermilk was then removed from the churner machine, and the butter grains were rinsed twice with cold, previously pasteurized drinking water (the time of each rinse was approximately 3 min at 8 °C). After the rinse water was removed, the butter was subjected to the kneading process, which removed the excess free rinse water, leading to the homogeneous dispersion of water in the fat phase. The experiment was conducted in duplicate.

The resulting butter samples were also stored at 25 °C and 5 °C for 21 days for further analysis. The water content and the degree of water dispersion were examined in the resulting butter. The butter samples were stored in portions of 70 g in sterile glass jars (tightly sealed and away from light). Microbiological (contaminating microflora) and physicochemical (water content, water distribution, pH value of the plasma, fat acidity, and oxidative stability) analysis of the butter samples was conducted at 0, 7, 14, and 21 days of sample storage.

#### 2.4.2. Microbiological Analysis of the Butter

The presence of contaminating microflora was measured after butter manufacture and during its refrigerated storage. The total number of *Enterobacteriaceae* was determined in VRBG medium (Merck) overlaid on the same medium and incubated at 37 °C for 24 h. The total numbers of molds and yeasts were determined in YGC medium (Merck), and the plates were incubated at 25 °C for 5 days. The result was given in colony-forming units of 1 mL (cfu/mL). The analysis was conducted in duplicate.

#### 2.4.3. Physicochemical Analysis of the Butter

Determining the water content in the butter consisted of establishing, by weight, the weight loss of the sample dried with properly prepared sand and calculating the percentage of water content in the product [22]. A weight of approximately 4 g of butter, mixed with sand, was dried in a laboratory dryer at 102 °C for 3 h, then cooled and redried for another hour at the same temperature. The drying process was conducted until there was no loss of weight of the dried sample. The analysis was conducted in duplicate.

The main principle of the method of water distribution in the butter is to apply indicator paper saturated with indicator to the freshly cut butter surface [23]. The analysis was performed using commercial indicator paper (Dysperwod, LABLACTA, Olsztyn, Poland) according to the manufacturer’s instructions. This method allows for the determination of whether the butter has been properly kneaded and the water droplets properly dispersed in the butter matrix. The indicator paper turns dark blue where it encounters water droplets. To determine the degree of water distribution, a point scale of 0–3 was used, according to which the products were classified using the criteria given in Table 1. The analysis was conducted in duplicate.

The pH value of the butter plasma. The test consisted of gently melting 40.0 g of butter sample in a water bath at 40 °C, then separating the aqueous phase from the fat phase using an MPW-350R centrifuge (MPW Med. Instruments, Warsaw, Poland) at 1100× *g* for 15 min at 40 °C. The fat separated on top was carefully collected for further analysis. The aqueous phase of the butter (butter plasma) was analyzed to measure the pH. The pH value of the butter plasma samples was measured at 25 °C using a CPO-505 pH meter (Elmetron) with a conventional electrode. The analysis was conducted in duplicate.

Determination of butter fat acidity. Ten grams of butter fat were weighed into Erlenmeyer flasks, then 25 mL of neutralized ethanol and 2 drops of 2% alcoholic phenolphthalein solution were added. Next, the content was heated to boiling and shaken vigorously, followed by titrations with 0.1 N NaOH until a slightly pink color was obtained that lasted for 30 sec. The analysis was conducted in duplicate. Fat acidity was expressed in degrees (°K) as a volume (mL) of sodium hydroxide solution of molar concentration (0.1 mol/dm^3^), spent for the neutralization of free acids in 10 g of product [24].

Oxidative stability. The butter samples were subjected to Rancimat accelerated oxidation test conditions, according to Gramza-Michalowska et al. [5], using a Rancimat-type 743 apparatus (Metrohm AG, Herisau, Switzerland). In the reaction vessel, a 3-g sample of butter fat phase was oxidized at 120 °C (air flow 20 dm^3^/h). The end of the induction period was characterized by a quick increase in water conductivity (due to the dissociation of volatile carboxylic acids). The analysis was conducted in duplicate.

### 2.5. Calculations and Statistical Analysis of the Results

Calculations of the mean and standard deviation values of the results were performed using an MS Office Excel 2016 spreadsheet. One-way or two-factor ANOVA analysis of variance was used for the statistical analysis of the obtained results in Statistica (version 13.3, StatSoft, Krakow, Poland). Comparisons of the significance of differences between the obtained results were made based on Tukey’s test (HSD) at the significance level α = 0.05.

## 3. Results and Discussion

Herbs and spices are natural ingredients, widely used as a food additive. The use of herbal extracts in dairy production can increase the variety of dairy-based products offered and may supply additional benefits. We aimed to study the effect of the addition of the whole extract of the herbs, but not the specific compounds isolated from them. We realized that the effect on the basis of specific compounds could be completely different from the use of the whole herbal extract. However, the activity of the antioxidant and antimicrobial components present in herbs makes it necessary to verify various aspects of their effects, with many technological challenges faced when developing dairy products enriched with herbs and spices. In this manuscript, we have focused on the effects of selected herbal extracts (as a whole), obtained with ethanol, and we hope that this will contribute to further research on individual compounds of these extracts showing antioxidant, antimicrobial or other interesting effect on the quality or stability of the dairy product.

### 3.1. Antimicrobial Activity of Herbal Extracts (Zone Inhibition)

From a dairy technology perspective, it is important not only to inhibit contaminating or pathogenic microflora, but also to ensure that the extracts do not affect the technical microflora.

The size of the zones of inhibition of bacterial growth with both the 68% ethanol and the DMSO aqueous solutions did not exceed 1 mm (Table 2). This means that the tested extracts of savory, basil, oregano, rosemary, and thyme did not inhibit the lactic fermentation bacteria included in the tested cultures at the set concentration. Most importantly, our earlier scientific studies confirm the results obtained in the present study [8,25]. Kozlowska et al. [25] conducted research on the effect of coriander essential oil on the growth of lactic acid bacterial strains. After analyzing the values of the average zones of growth inhibition of the tested strains of lactobacilli, it was found that they varied depending on the type and concentration of oil used and the bacterial strain used. As the concentration of added coriander essential oils increased, a larger zone of growth inhibition of the tested *Lactobacillus* bacteria was observed. This may favor the use of these oils for *Lactobacillus* products. Furthermore, a study by Piasecka-Jóźwiak et al. [26] showed that thyme oil had the strongest negative effect on all the bacterial strains they evaluated. A slight increase in the concentration of thyme oil to 0.1% in the MRS agar resulted in a decrease in the number of bacterial cells below the baseline level. Of all the LAB (lactic acid bacteria) assessed, only *Lactobacillus lactis* subsp. *lactis* KKP 835 was characterized by its ability to grow under these conditions. The highest concentration of thyme oil that did not affect bacterial growth for this strain was 0.1% (*v*/*v*).

It transpires that, in terms of the effect of extracts on starter culture microflora, plant extracts show high selectivity. For example, an extract from the leaves of densiflora pine (*Pinus densiflora*) showed activity against *Clostridium perfringens*, *Staphylococcus aureus*, and *Escherichia coli*, without inhibiting the growth of beneficial microflora such as *Lactobacillus* and *Bifidobacterium* [3,11,25]. Diniz do Nascimento et al. [27] showed the inhibitory effect of the essential oils of clove, cinnamon, and rosemary on some species of *Lactobacillus*, but Ali et al. [28] already observed no inhibitory effect of rosemary on the growth of *L. plantarum*. However, Saguibo and Elegado [29] noticed the resistance of the probiotic lactobacilli against some plant extracts, including avocado (*Persea americana* Mill.) and malunggay (*Moringa oleifera* Lam.) leaves.

### 3.2. Change in pH Value during Cream Fermentation

The resulting pH measurements taken during the 6-h fermentation of the cream are shown in Figure 1. The results of the pH measurement taken during the 6-h fermentation of the cream with the YC-X16 culture are shown in Figure 1a and the initial pH of the cream averaged 6.69 ± 0.13. Meanwhile, the results of the pH measurement performed during the 6-h fermentation of the cream with the YO-MIX 207 culture are shown in Figure 1b and the initial pH of the cream averaged 6.69 ± 0.12. The samples initially did not deviate from the original pH, regardless of the additive and the type of herbal extract used.

From the data presented in Figure 1, it can be seen that the control sample, which did not hold any of the tested extracts, statistically significantly differed from the other samples with the addition of the savory, rosemary, oregano, basil, and thyme extracts, which are determined as constituting one homogeneous group. Statistical analysis distinguished two homogeneous groups, one of which included the cream samples containing the oregano, basil, thyme, rosemary, and savory extracts, and the other the control sample of cream. This means that the addition of the tested extracts at a fixed level did not inhibit the fermentation of the cream, and their presence contributed to a shorter fermentation course. In turn, grouping the results of the two-factor analysis of variance according to fermentation time resulted in the determination of six homogeneous groups. While later measurements were taken, significant differences in pH changes were noted only after the 5th hour as fermentation slowed down, and the results were not significantly different from each other.

Comparable results were obtained by Amirdivani and Baba [30], who showed that herbal extracts improved milk fermentation through yogurt bacteria and increased the acidity of yogurts. Khelif et al. [31] conducted a study to investigate the effect of thyme extract on LAB growth and concluded that a 2% and 4% addition of thyme extract can be added to yogurts without the risk of deterioration in physicochemical quality or inhibition of lactic bacteria. Bakrm and Salihin [32] found that adding aqueous extract of Ceylon cinnamon and common garlic to goat’s milk, cow’s milk, and camel’s milk had no negative effect on acidification through fermentation. Other studies report that the addition of moringa, peppermint, fennel, or common basil extract to yogurt reduces fermentation time by increasing the growth of yogurt cultures [33].

### 3.3. Counts of Starter Culture Bacteria during Cream Fermentation

The results presented in Figure 2 show the survival of the microflora during the fermentation process of cream via the YC-X16 starter culture (Figure 2a) and the YO-MIX 207 starter culture (Figure 2b), and with addition of tested extracts. From the presented data, it can be seen that the addition of almost all of the tested herbal extracts to the cream had a significant (*p* < 0.05) effect on the cell counts of bacteria of the *S. thermophilus* species, as well as a selective influence on the genus *Lactobacillus*, bifidobacteria, and the *L. acidophilus* species. The statistical analysis confirmed these observations (*p* < 0.05). When comparing these results with the pH measurements, we found no clear correlations. For bacterial cell counts, the addition of herbal extracts sometimes stimulated a higher number of cells than in the control sample (e.g., *L. acidophilus* in the cream with the YO-MIX 207 culture), and in some samples, the control sample contained a higher bacterial counts than the other cream samples (e.g., lactobacilli or *S. thermophilus* in the cream with the YC-X16 culture, and bifidobacteria in the cream with the YO-MIX 207 culture). In addition, by comparing the changes in lactobacilli counts in the cream samples with the YC-X16 culture and the YO-MIX 207 culture, it can be concluded that the effect of the addition of herbal extracts on lactic acid bacteria is dependent on the choice of bacterial strains. These changes were not reflected in the pH changes described above.

Research reports on the effect of plant extracts on the survival of yogurt cultures during fermentation show major differences in results. They vary depending on the bacteria, the composition of the starter culture, the type and amount of extract added, its concentration, and its sensitivity to low pH, among other factors. Some of the data from the literature overlap with the results of the present study, but there are also many discrepancies that require further analysis [34,35,36,37,38,39]. First, the bacterial cell counts in yogurt often decreases below the recommended level (more than 6.0 log cfu/g) [40] due to low pH, high oxygen concentration, increased redox potential, and increased hydrogen peroxide concentration [41,42,43,44,45], making it even more difficult to correctly interpret the effect of the herbal extract addition on the counts of lactic acid bacteria. During fermentation, intensive transformations of milk components take place. Furthermore, the higher buffering capacity of fermented milk beverages can counteract the negative effects of an acidic environment on the counts of lactic acid bacteria and bifidobacteria [38]. The results coinciding with ours were obtained by Arslan et al. [34], who found that yogurt samples with basil and coriander essential oils had lower lactic bacteria counts than the control samples, and, in contrast, the samples with savory oil showed higher lactic bacteria counts than the control samples. Marhamatizadeh et al. [37], on the other hand, showed a positive relationship between growth in *L. acidophilus* and *B. bifidum* and the level of olive leaf extract addition. Moreover, Joung et al. [39], studying the properties of herbal yogurt with traditional Korean lotus nut plant extracts, observed an increase in the viability of starter culture bacteria. In contrast, Behrad et al. [36] reported that yogurt mixed with cinnamon and licorice herbs had lower numbers of *L. bulgaricus* and *S. thermophilus* live cells compared with the control yogurt.

### 3.4. Counts of Contaminating Microflora during Cream Fermentation

The results obtained in this study indicated that molds, yeast, and *Enterobacteriaceae* were found to be absent in all samples during the fermentation of the cream.

### 3.5. Counts of Starter Culture Bacteria during Cream Storage

The aim of the study is to investigate the effect of adding selected herbal extracts on the viability of the starter culture bacteria, since they were present as viable cells in the final product, which determined the dietary value of the fermented flavored cream. It was interesting to see if the addition of selected herbal extracts affected the counts of viable cells of starter culture bacteria during fermented flavored cream storage. According to the data presented in Figure 3, the addition of each of the tested herbal extracts to the cream had a different effect on the bacterial cell viability during the 21 days of storage at 5 °C. Statistical analysis also showed that the refrigerated storage conditions of the cream with the herbal extract additives caused significant (*p* < 0.05) changes in the viability of lactic acid bacteria cells. In general, the greatest changes and the most statistically significant (*p* < 0.05) impact of the herbal extract additives were observed for the lactobacilli counts in the cream samples fermented with the YC-X16 culture during the 21 days of storage at 5 °C (Figure 3a). The largest statistically significant changes in lactobacilli counts during the 21 days of storage at 5 °C were observed in the savory extract cream samples fermented with the YC-X16 culture. An analysis of variance showed that there were significant changes in the number of cells of the microorganisms in question during successive days of refrigerated storage, relative to the number of cells obtained immediately after the fermentation of the cream.

The stimulating effect of plant extract additives was observed by Marhamatizadeh et al. [37], who studied the effect of olive leaf extract on the growth and viability of *L. acidophilus* and *B. bifidum* in probiotic milk and yogurt after 21 days of refrigerated storage. The researchers used a 2%, 4%, and 6% addition of olive leaf extract. The results showed that the counts of *L. acidophilus* and *B. bifidum* were higher than in the control sample when olive leaf extract was used, and there was a positive relationship between bacterial growth and the level of extract addition. In contrast, according to Arslan et al. [34], the counts of live lactic acid bacteria cells in yogurt samples with basil and coriander essential oils was lower than in the control samples, while in yogurt samples with savory oil, it was higher. Our research confirms the observations made by Joung et al. [39], who showed that the herbal extract and storage time affected several properties of the yogurt, changing the viability of the starter culture. Studies by other authors [35,36,38] show that the effect of adding extracts on yogurt culture viability depends on the type of plant from which the extract is obtained, as well as on the concentrations of phytochemicals present in the extracts used.

### 3.6. Counts of Contaminating Microflora during Cream Storage

Molds, yeast, and *Enterobacteriaceae* bacteria were all found to be absent in the fermented cream samples until the conclusion of the storage period at 5 °C. This indicates that the observed effects of the herbal extracts used were only related to the starter microflora.

Typically, butter should contain up to 1000 live cells of non-pathogenic microorganisms in 1 g [46], although this may be a higher value for butter derived from fermented cream [47]. The use of a step in which butter lumps are rinsed with microbiologically pure water means that butter carries a low microbial load [46]. In addition, proper hygienic production conditions, the right level of water content, and a high degree of water dispersion mean that butter is not a favorable environment for the growth of microorganisms [46,48].

### 3.7. Water Content in the Butter

The effect of the addition of the tested plant extracts to the fermented cream was also studied in the resulting butter (the butter was obtained from freshly fermented cream without refrigerated long-term storage). The measured water content was in the range of 15.9–16.0% ± 0.1, regardless of the addition of the herbal extract or the starter culture used to ferment the cream. According to the Codex Alimentarius [49], the water content in butter must not be greater than 16%. The results of the present study indicated that the obtained butter samples met these conditions.

### 3.8. Degree of Water Distribution in the Butter

The tested butter samples received the maximum number of points (class 3 according to Table 1) when determining the degree of water dispersion. No spots present on the indicator paper were noted on the fresh surface of any of the butter samples, regardless of the addition of the herbal extract or the starter culture used to ferment the cream.

The degree of water dispersion is important from a microbiological perspective [50]. The water phase present in the butter samples allowed optimum conditions for the proliferation of this microflora and the deterioration in the organoleptic qualities of the product, especially at the long-term storage stage. Therefore, the greater the degree of water dispersion, the more difficult the development of undesirable microflora.

### 3.9. pH Value of the Butter Plasma

An important stage in the production of butter, which subsequently affects the pH of the aqueous phase, is the biological maturation of the cream, i.e., its fermentation [51]. Moreover, the transformations in the aqueous phase of butter are most easily seen by observing changes in the pH of the plasma. Table 3 depicts that the plasma acidity of the analyzed butter samples obtained similar values to each other, regardless of the addition of the herbal extract or the starter culture used to ferment the cream. This means that the herbal extract additives used had no statistical effect (*p* < 0.05) on the plasma pH of the butter samples obtained. In addition, such changes were not observed in the butter samples stored at different temperatures: refrigerated (5 °C) and room (25 °C).

### 3.10. Determination of Butter Fat Acidity

From the data presented in Table 4, it can be seen that the addition of herbal extracts had no statistically significant effect (*p* < 0.05) on the acidity of the milk fat. The values found for this acidity were indistinguishable, regardless of the type of herbal extract added, the cream fermentation culture used, or the storage temperature of the butter samples. The only samples whose acidity changed unfavorably during storage were the control samples of butter (without added herbal extracts) kept at 25 °C for 21 days.

In determining the storage stability of butter, the acidity of butter fat is an expression of the amount of free fatty acid and the degree of lipolysis [24,52,53]. Our results were similar to those reported by Trawińska [54], who evaluated that the acidity of butter fat is a result of how the product is stored. A study by Trawińska [54] examined how storage at −8 °C, 0 °C, and 20 °C affected the acidity of butter fat. Negative temperatures had no effect on the acidity level of butter. While at 20 °C, it was shown that the acidity up to day 14 was at a similar level, and then increased. This proves that storage temperature is a key factor affecting the overall quality characteristics of butter. The cited researcher showed that negative quality characteristics can be observed when butter is kept at 20 °C for long periods. In our study, statistical analysis demonstrated that the use of herbal extracts had a significant effect on the fat acidity of butter compared with the control samples.

### 3.11. Counts of Contaminating Microflora

Molds, yeast, and *Enterobacteriaceae* were all found to be absent in the obtained butter samples with the herbal extract additives until the conclusion of the storage period at 5 °C. This proves the high standard of hygiene in the production of the butter, as well as the positive impact of the used starter cultures on the microbiological quality of the resulting products [55].

### 3.12. Oxidative Stability

The effect of the addition of the herbal extracts was only noticed when analyzing lipid stability under Rancimat test conditions. This analysis allowed an evaluation of the induction period based on the increase in water conductivity caused by the oxidation process [56,57]. Table 5 shows the duration of the oxidative stability measurement of the tested samples of butter fat obtained from cream with the addition of the tested herbal extracts.

Even immediately after butter manufacture, its samples were characterized by varying induction periods. Statistically significant effect (*p* < 0.05) on the extension of its value had the rosemary extract and thyme extract additions, in the other cases of additives. The values of the induction period did not differ from the value obtained for the control sample, regardless of the type of starter culture used for the fermentation of the cream intended for butter production.

The oxidative stability of milk fat was also measured after 21 days of storage at 5 °C and 25 °C to compare the effect of added plant extracts on the shelf life of butter over time. After storage at 25 °C for 21 days, as in the case of the measurement immediately after butter manufacture, the longest induction time was characterized by the milk fat samples with the rosemary and thyme extracts. Based on an analysis of variance, it was found that the other induction time results were not statistically significantly different from each other (*p* < 0.05). Analogous results were obtained for the samples stored at 5 °C for 21 days, which proved the positive effect of the addition of rosemary and thyme on the period of milk fat induction.

The storage conditions also had a significant impact on butter stability [53,58]. As expected, the oxidative stability of the milk fat obtained was higher for the samples of butter kept at 6 °C compared with those kept at 25 °C.

Spices and herbs are known to have enormous potential as natural antioxidants in food, and the general principle of antioxidants is their reaction with oxidizing agents—free radicals. Kozowska et al. [59] showed that herbs from the Lamiaceae family (including thyme, oregano, rosemary, lemon balm, savory, hyssop, sage, narrow-leaved lavender and clary sage) have potent antioxidant activity, mainly due to the phenolic compounds present in them. These include eugenol, carvacrol and thymol. This is supported by the study by Bandoniene et al. [60]. Rosemary extract (*Rosmarinus officinalis*) is one of the most used extracts for this purpose. Its strong antioxidant properties are attributed to its high phenolic component content [8,59]. These properties may even be many times stronger than those found for synthetic antioxidants such as BHA (butylated hydroxyanisole) and BHT (butylated hydroxytoluene) [61,62,63,64]. Médici Veronezi et al. [65] showed an increase in the oxidative stability of thermo-oxidized soybean oil using the example of basil ethanol extract. It is worth noting that the same compounds are responsible for the bactericidal or bacteriostatic properties of herbal extracts. In addition, Kozlowska et al. [59] found that aqueous ethanolic extracts of thyme, rosemary and sage contained slightly higher levels of phenolic compounds compared to aqueous methanolic extracts.

Some authors have also reported the antimicrobial and antioxidant effects of extracts and essential oils of herbs on butter [66,67]. The results of Ayar et al. [68] indicate that methanolic extracts of sage, rosemary, and oregano have a significant effect on butter stability, especially at the 0.05% level. The most effective antioxidants used were sage extract, a sage-rosemary combination, and rosemary extract. Oregano extract and its combinations with rosemary and sage extracts have shown pro-oxidant activity. In addition, Gramza-Michalowska et al. [5] observed that rosemary extract was also an active antioxidant, allowing lipid stabilization two times longer compared with butter control samples. The results of our research are consistent with the results of Gramza-Michalowska et al. [5]. Thus, natural spices and their combinations can be used to increase the oxidative stability of butter.

To conclude, herbs and spices are natural ingredients widely used as food additives. The addition of herbs and spices to dairy products for health benefits should meet requirements in terms of safety, performance, price, and release to avoid any side effects. The use of carefully selected extracts will create products with natural health-promoting properties, with the effect of stabilizing quality parameters.

The main reason for the shelf-life limit of butter is changes in the lipid fraction. They progress as butter is stored and are the result of oxidation and the lipolytic activity of microbial and milk-derived enzymes [5,52]. Regarding the experimental results obtained, it can be concluded that the tested solutions of the savory, basil, oregano, rosemary, and thyme extracts do not inhibit the growth of the lactic fermentation bacteria included in the industrial starter cultures.

To summarize, the present study indicates the strong antioxidant activity of the selected examined herbal extracts in lipid systems. However, Kozłowska et al. [8] found no relationship between the phenolic content of spice extracts and their ability to inhibit LAB growth. Of the plant extracts evaluated, the rosemary and thyme extracts had a significant beneficial effect on the oxidative stability of butter. The other plant extracts had no statistically significant effect on the oxidative stability of butter. No significant effect of storage temperature was observed on the obtained results. Therefore, plant extracts can be successfully used as a functional additive to protect the lipid fraction of fermented foods.

## 4. Conclusions

Spices and herbs are known to have enormous potential as natural antioxidants in foods. In conclusion, rosemary extract (*Rosmarinus officinalis* L.) is one of the most used extracts for this purpose. Its strong antioxidant properties are attributed to its high phenolic component content. The presence of extracts of savory, basil, oregano, rosemary, and thyme alters the course of fermentation by lactic bacteria. However, the addition of savory, basil, oregano, rosemary, or thyme extracts in the concentration range used in this work does not affect the cell counts of starter cultures under the conditions of cream fermentation and the refrigerated storage of cream.

Of the plant extracts evaluated, the rosemary and thyme extracts had a significant beneficial effect on the oxidative stability of butter. No significant effect of temperature was observed on the obtained results. The other plant extracts had no statistically significant effect on the oxidative stability of butter. Therefore, plant extracts can be successfully used as a functional additive to protect the lipid fraction of fermented foods. The research results presented do not exhaust the research or application area. When developing new products, sensory evaluation is extremely important. Such an analysis was not carried out in the present study. In the future, these investigations must be continued with regard to the influence of the addition of extracts on the storage stability of cream, the sensory evaluation of flavored cream and flavored butter, and the nutritional parameters of these products. However, more detailed studies are also needed to determine the effects of fresh or dried spices added directly to dishes, even though such products already exist on the market and in traditional cuisines [64].

## Figures and Tables

**Figure 1 foods-12-00471-f001:**
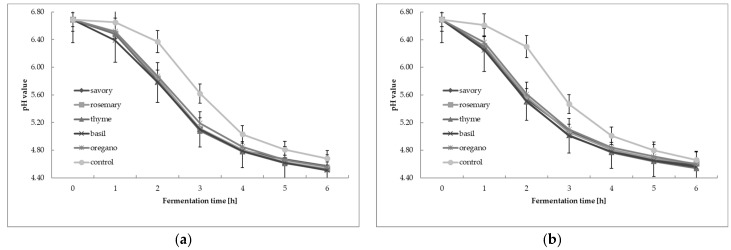
Change in pH value during fermentation of cream with: (**a**) YC-X16 culture; (**b**) YO-MIX 207 culture with the addition of tested extracts. Mean of three replications ± standard deviation (SD).

**Figure 2 foods-12-00471-f002:**
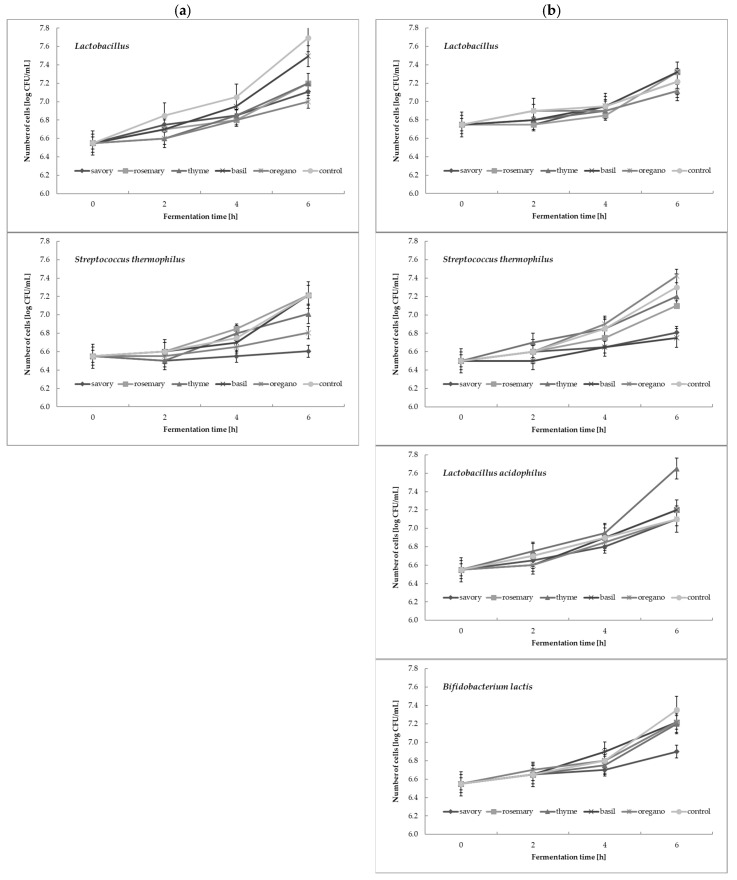
The counts of microflora during fermentation of cream with: (**a**) YC-X16 culture; (**b**) YO-MIX 207 culture with addition of tested extracts. Mean of three replications ± standard deviation (SD).

**Figure 3 foods-12-00471-f003:**
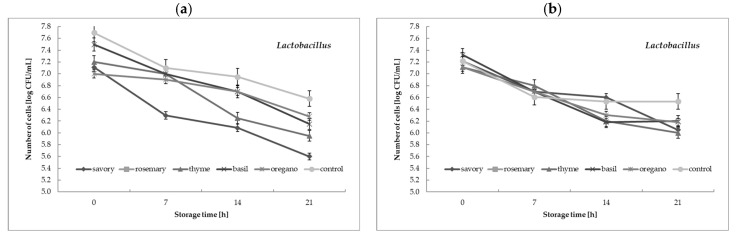
The survival of microflora in the cream fermented with: (**a**) YC-X16 culture; (**b**) YO-MIX 207 culture during 21 days of storage at 5 °C. Mean of three replications ± standard deviation (SD).

**Table 1 foods-12-00471-t001:** Classification of butter by the degree of water distribution.

Verbal Definition of Water Distribution in the Butter	The Size (Diameter) and Density of the Spots Present on the Indicator Paper	Class
Very bad	Diameter 3–8 mm occurring densely (occupy about 20% of the paper surface)	0
Bad	Diameter 1–3 mm occurring moderately densely (occupy about 10% of the paper surface)	1
Sufficient	Diameter 0.3–1 mm occurring rarely (occupy about 5% of the paper surface)	2
Good	No spots	3

**Table 2 foods-12-00471-t002:** Antimicrobial activity of spice extracts (zone inhibition).

Extract of	In DMSO	In 68% Ethanol Solution	Extract of	In DMSO	In 68% Ethanol Solution
	The Size of the Inhibition Zone [mm]			The Size of the Inhibition Zone [mm]	
YC-X16 Culture			YO-MIX 207 Culture		
*Streptococcus thermophilus*			*Streptococcus thermophilus*		
savory	0.0 ± 0.1 *	0.0 ± 0.0	savory	0.1 ± 0.0	0.0 ± 0.0
basil	0.1 ± 0.0	0.1 ± 0.0	basil	0.0 ± 0.0	0.1 ± 0.0
thyme	0.0 ± 0.1	0.0 ± 0.0	thyme	0.1 ± 0.0	0.1 ± 0.0
rosemary	0.0 ± 0.0	0.1 ± 0.0	rosemary	0.1 ± 0.0	0.1 ± 0.0
oregano	0.0 ± 0.1	0.0 ± 0.0	oregano	0.1 ± 0.0	0.0 ± 0.0
*Lactobacillus*			*Lactobacillus*		
savory	0.1 ± 0.0	0.0 ± 0.1	savory	0.1 ± 0.0	0.0 ± 0.0
basil	0.0 ± 0.1	0.0 ± 0.1	basil	0.0 ± 0.0	0.1 ± 0.0
thyme	0.0 ± 0.0	0.0 ± 0.0	thyme	0.1 ± 0.0	0.1 ± 0.0
rosemary	0.1 ± 0.0	0.0 ± 0.1	rosemary	0.1 ± 0.0	0.0 ± 0.1
oregano	0.1 ± 0.0	0.0 ± 0.1	oregano	0.0 ± 0.1	0.1 ± 0.0
			*Lactobacillus acidophilus*		
			savory	0.0 ± 0.1	0.1 ± 0.0
			basil	0.0 ± 0.1	0.0 ± 0.0
			thyme	0.0 ± 0.0	0.1 ± 0.0
			rosemary	0.0 ± 0.1	0.1 ± 0.0
			oregano	0.0 ± 0.1	0.0 ± 0.1
			*Bifidobacterium lactis*		
			savory	0.0 ± 0.1	0.1 ± 0.0
			basil	0.1 ± 0.0	0.0 ± 0.0
			thyme	0.0 ± 0.1	0.1 ± 0.0
			rosemary	0.0 ± 0.0	0.1 ± 0.0
			oregano	0.0 ± 0.1	0.0 ± 0.1

* Mean of four replications ± standard deviation (SD). In each case, there was no statistically significant effect of herbal extracts on the size of the inhibition zone.

**Table 3 foods-12-00471-t003:** The results of the plasma acidity measurement of butter with the addition of the tested plant extracts after 21 days of storage at 25 °C and 5 °C.

**Butter with YC-X16 Culture**			
**Extract**	**pH Value**		
	**Immediately after Preparation**	**Storage Conditions** **21 Days/25 °C**	**Storage Conditions** **21 Days/5 °C**
savory	4.6 ± 0.1 ^a^	4.5 ± 0.2 ^a^	4.5 ± 0.1 ^a^
basil	4.4 ± 0.1 ^a^	4.5 ± 0.1 ^a^	4.4 ± 0.2 ^a^
thyme	4.5 ± 0.1 ^a^	4.5 ± 0.4 ^a^	4.5 ± 0.1 ^a^
rosemary	4.7 ± 0.1 ^a^	4.4 ± 0.1 ^a^	4.6 ± 0.1 ^a^
oregano	4.5 ± 0.1 ^a^	4.4 ± 0.2 ^a^	4.5 ± 0.1 ^a^
control	4.6 ± 0.1 ^a^	4.5 ± 0.1 ^a^	4.5 ± 0.2 ^a^
**Butter with YO-MIX 207 Culture**			
**Extract**	**pH Value**		
	**Immediately after Preparation**	**Storage Conditions** **21 Days/25 °C**	**Storage Conditions** **21 Days/5 °C**
savory	4.6 ± 0.1 ^a^	4.5 ± 0.2 ^a^	4.5 ± 0.1 ^a^
basil	4.4 ± 0.1 ^a^	4.3 ± 0.2 ^a^	4.4 ± 0.1 ^a^
thyme	4.5 ± 0.1 ^a^	4.3 ± 0.2 ^a^	4.5 ± 0.1 ^a^
rosemary	4.6 ± 0.1 ^a^	4.6 ± 0.2 ^a^	4.6 ± 0.1 ^a^
oregano	4.5 ± 0.1 ^a^	4.4 ± 0.1 ^a^	4.5 ± 0.2 ^a^
control	4.6 ± 0.1 ^a^	4.6 ± 0.2 ^a^	4.6 ± 0.1 ^a^

^a^ Mean of three replications ± standard deviation (SD). Means with the same lowercase letter within the same row indicate no significant difference at the significance level of 0.05.

**Table 4 foods-12-00471-t004:** The results of the measurement of the fat acidity of butter with the addition of the tested herbal extracts after 21 days of storage at 25 °C and 5 °C.

**Milk Fat from Butter with YC-X16 Culture**			
**Extract**	**Degrees of Acidity**		
	**Immediately after Preparation**	**Storage Conditions** **21 Days/25 °C**	**Storage Conditions** **21 Days/5 °C**
savory	1.7 ±0.1 ^a^	1.7 ±0.1 ^a^	1.6 ±0.1 ^a^
basil	1.8 ±0.1 ^a^	1.9 ±0.1 ^a^	1.9 ±0.1 ^a^
thyme	1.8 ±0.0 ^a^	1.9 ±0.1 ^a^	1.9 ±0.1 ^a^
rosemary	1.5 ±0.1 ^a^	1.5 ±0.1 ^a^	1.5 ±0.1 ^a^
oregano	1.8 ±0.1 ^a^	1.8 ±0.1 ^a^	1.7 ±0.1 ^a^
control	1.7 ±0.1 ^a^	2.5 ±0.1 ^b^	1.9 ±0.1 ^a^
**Milk Fat from Butter with YO-MIX 207 Culture**			
**Extract**	**Degrees of Acidity**		
	**Immediately after Preparation**	**Storage Conditions** **21 Days/25 °C**	**Storage Conditions** **21 Days/5 °C**
savory	1.6 ±0.1 ^a^	1.7 ±0.1 ^a^	1.6 ±0.1 ^a^
basil	1.7 ±0.1 ^a^	1.9 ±0.1 ^a^	1.8 ±0.1 ^a^
thyme	1.6 ±0.1 ^a^	1.6 ±0.1 ^a^	1.5 ±0.1 ^a^
rosemary	1.8 ±0.1 ^a^	1.9 ±0.1 ^a^	1.8 ±0.1 ^a^
oregano	1.5 ±0.1 ^a^	1.6 ±0.1 ^a^	1.5 ±0.1 ^a^
control	1.8 ±0.1 ^a^	2.4 ±0.1 ^b^	1.8 ±0.1 ^a^

^a, b^ Mean of three replications ± standard deviation (SD). Means with the same lowercase letter within the same row indicate no significant difference at the significance level of 0.05.

**Table 5 foods-12-00471-t005:** The results of the oxidative stability measurement of milk fat (from butter with the addition of tested plant extracts) after 21 days of storage at 25 °C and 5 °C.

**Milk Fat from Butter with YC-X16 Culture**			
**Extract**	**Measurement Duration [h]**		
	**Immediately after Preparation**	**Storage Conditions** **21 Days/25 °C**	**Storage Conditions** **21 Days/6 °C**
savory	9.01 ± 0.37 ^a^	10.05 ± 1.28 ^a^	8.24 ± 0.98 ^a^
rosemary	39.99 ± 0.41 ^c^	37.04 ± 1.44 ^c^	74.38 ± 1.57 ^d^
thyme	15.99 ± 2.21 ^b^	17.93 ± 1.48 ^b^	18.93 ± 1.48 ^b^
basil	7.82 ± 0.81 ^a^	8.24 ± 0.38 ^a^	6.50 ± 0.86 ^a^
oregano	10.91 ± 0.10 ^a^	9.48 ± 0.58 ^a^	10.41 ± 1.02 ^a^
control	9.20 ± 0.31 ^a^	7.26 ± 1.10 ^a^	8.02 ± 0.95 ^a^
**Milk Fat from Butter with YO-MIX 207 Culture**			
**Extract**	**Measurement Duration [h]**		
	**Immediately after Preparation**	**Storage Conditions** **21 Days/25 °C**	**Storage Conditions** **21 Days/6 °C**
savory	8.95 ± 1.34 ^a^	10.07 ± 1.30 ^a^	8.31 ± 0.98 ^a^
rosemary	39.50 ± 2.12 ^c^	36.65 ± 0.78 ^c^	77.35 ± 1.63 ^d^
thyme	17.22 ± 0.92 ^b^	17.98 ± 1.52 ^b^	20.25 ± 1.58 ^b^
basil	8.55 ± 0.59 ^a^	8.26 ± 1.05 ^a^	6.63 ± 0.88 ^a^
oregano	10.80 ± 0.35 ^a^	9.50 ± 0.59 ^a^	10.79 ± 1.01 ^a^
control	9.55 ± 0.74 ^a^	6.77 ± 0.38 ^a^	7.06 ± 0.91 ^a^

^a–d^ Mean of three replications ± standard deviation (SD). Means with different lowercase letters within the same row indicate a significant difference at the significance level of 0.05.

## Data Availability

Data is contained within the article.

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
