# Peer review of "Effect of the Addition of Selected Herbal Extracts on the Quality Characteristics of Flavored Cream and Butter"

_foods, 2023, doi:10.3390/foods12030471_

Round 1

Reviewer 1 Report

The manuscript by Ziarno et al. reports about the addiction of herbal extracts in fermented flavored cream and the resulting flavored butter, in terms of effects on the microbiota consortia, butter plasma pH, butter fat acidity, and oxidative stability of the obtained final product.

The study appear interesting with appropriate methods of investigations and analysis.

Some minor concerns are summarized below and must be revised. The authors should respond to these comments.

Minor revisions.

- lines 63-73: it is not clear the amount of herbal extract considered by the authors in their experiments. Did this extract correspond to the mentioned 10g of dried herb leaves (lines 67-68) for each performed experiments? Please clarify

- lines 90-91: why authors choose precisely 0,001% of herbal extract solution in the two solvent for this test? Please clarify

- paragraphs in Material and Methods section could be combined in one paragraph or precisely:  2.3.1 and 2.3.2, 2.3.3 and 2.3.4, 2.4.1 and 2.4.2, 2.4.3 and 2.4.4 and 2.4.5. Consequently, corresponding Results paragraphs could be combined as well.

- as general consideration, Results section seems to be organized as a “Results and Discussion” ones. Many paragraphs have comments on the reported results with references (i.e. from lines 234 to 258; from lines 282 to 291; from lines 311 to 333; from lines 354 to 368; from lines 376 to 382, from lines 395 to 399; from lines 421 to 432; from lines 468 to 484). Results section should reports about the obtained results bearing information about the emerged significance with respect the statistics considered, so comments should be included in the Discussion section. Please reorganize Results and Discussion sections accordingly.

Author Response

  1. Authors’ Responses

The manuscript by Ziarno et al. reports about the addiction of herbal extracts in fermented flavored cream and the resulting flavored butter, in terms of effects on the microbiota consortia, butter plasma pH, butter fat acidity, and oxidative stability of the obtained final product.

The study appear interesting with appropriate methods of investigations and analysis.

Some minor concerns are summarized below and must be revised. The authors should respond to these comments.

Authors’ Response: We thank the Reviewer for the generous comments and suggestions on the manuscript and we have edited the manuscript to address these concerns. We have revised and edited the manuscript according to the reviewer’ comments and we believed that the manuscript is now suitable for publication.

Minor revisions.

- lines 63-73: it is not clear the amount of herbal extract considered by the authors in their experiments. Did this extract correspond to the mentioned 10g of dried herb leaves (lines 67-68) for each performed experiments? Please clarify

Authors’ Response: Thank you for your time and kind consideration. We were only describing the general principle of obtaining the herbal extracts, the amounts used did not correspond to the doses used in the experiments. For a clearer understanding of the methodology, we have rewritten the methodological description for obtaining herbal extracts.

- lines 90-91: why authors choose precisely 0,001% of herbal extract solution in the two solvent for this test? Please clarify

Authors’ Response: Thank you for that comment. We chose this level of additive suggesting the regulations given in Commission Regulation (EU) No 1129/2011 of 11 November 2011 amending Annex II to Regulation (EC) No 1333/2008 of the European Parliament and of the Council by establishing a Union list of food additives (OJ L 295, 12.11.2011, p.1) regarding the use of extracts of rosemary for food. In addition, we previously did experiments with higher doses even to the level of 0.1% (unpublished data), which were not successful considering aspects of butter flavor and odor.

- paragraphs in Material and Methods section could be combined in one paragraph or precisely:  2.3.1 and 2.3.2, 2.3.3 and 2.3.4, 2.4.1 and 2.4.2, 2.4.3 and 2.4.4 and 2.4.5. Consequently, corresponding Results paragraphs could be combined as well.

Authors’ Response: Thank you for these suggestions. We have revised the methods described and re-grouped them according to the suggestions.

- as general consideration, Results section seems to be organized as a “Results and Discussion” ones. Many paragraphs have comments on the reported results with references (i.e. from lines 234 to 258; from lines 282 to 291; from lines 311 to 333; from lines 354 to 368; from lines 376 to 382, from lines 395 to 399; from lines 421 to 432; from lines 468 to 484). Results section should reports about the obtained results bearing information about the emerged significance with respect the statistics considered, so comments should be included in the Discussion section. Please reorganize Results and Discussion sections accordingly.

Authors’ Response: Thank you for that comment. Of course, we acknowledge the point. Inadvertently, we did not title the sections with the correct names, hence the incorrect chapter structure of the manuscript. We have made the correction as suggested by the reviewer.

Reviewer 2 Report

1. The species of the selected herbs should be verified via botanist etc based on the voucher no. (if possible).

2. Different plant species might content different A.i. for specific purposes. I don't see any of those specific A.i. used for the additives mechanism effect on the diary product. 

3. Please specify what is your interested compounds compound in the extract that need to be highlighted? e.g. phenolic? Alkaloid? Be specific.

4. I don't have any issues on the further experimental works as a result of your extract usage. But the extract itself does not reflect the specific compound that will give the effect to milk product based. Need very thorough info on the extracts. It seems that the extract preparation of different plant species were the same. 

Author Response

  1. Authors’ Responses

Dear Reviewer, thank you for your careful comments and your valuable time.

We have studied your comments carefully and have made revision which highlighted in the text. Now, we hope that the revision is acceptable for publication.

The revised parts according to the comments as follows:

  1. The species of the selected herbs should be verified via botanist etc based on the voucher no. (if possible).

Authors’ Response: Thank you for your time and kind consideration. According to the American Botanical Council, a voucher specimen is a botanical reference material associated with a specific lot or batch of biomass, and, as such, serves to document the authenticity of the bulk material (DOI: 10.1007/s00216-007-1405-x). In botanical practice, a voucher specimen includes all available aboveground portions of a representative specimen of a particular plant population, properly dried and affixed to a herbarium sheet. Voucher specimens are critical in confirming and documenting the identity of research and commercial botanicals. In the case of our research study, the raw material included extracts of dried leaves of popular herbs commercially available in the grocery shops. For this reason, voucher no. are not necessary, in our opinion. Therefore, we have given the systematic classification created by Linnaeus for the individual herbs: thyme (Thymus vulgaris L.), oregano (Origanum vulgare L.), savory (Satureja hortensis L.), rosemary (Rosmarinus officinalis L.), and basil (Ocimum basilicum L.).

  1. Different plant species might content different A.i. for specific purposes. I don't see any of those specific A.i. used for the additives mechanism effect on the diary product. 

Authors’ Response: We are aware that different herbs show different activity index (AI) values relevant to different effects. For this reason, in this research we did not want to get into the detailed chemical composition of the individual herbal extracts, but only to treat them as whole multicomponent systems and so test them for their effects on example dairy products. We agree with the Reviewer that not all herbal extracts show similar AI. This was confirmed by our study described in this manuscript. Our study should be regarded as an initiation for further research studies, and in our opinion, the results obtained seem to indicate which herbal extracts are worthy of further interest.

  1. Please specify what is your interested compounds compound in the extract that need to be highlighted? e.g. phenolic? Alkaloid? Be specific.

Authors’ Response: Thank you very much for this comment. We understand now that our description in the manuscript did not correctly reflect our research intentions. We are in most agreement with the Reviewer that the text of a scientific manuscript must be clearly understood by all readers. In this research, we were interested in whole extracts as a natural matrix of different components, which we did not intend to single out at this point, but only to recognise as a natural system characteristic of specific herbs. To make this clearer in the manuscript, we have added a few sentences to the aim of the study, the methodological description and the discussion of the results.

  1. I don't have any issues on the further experimental works as a result of your extract usage. But the extract itself does not reflect the specific compound that will give the effect to milk product based. Need very thorough info on the extracts. It seems that the extract preparation of different plant species were the same. 

Authors’ Response: Yes, the preparation of the extract from the different plants was the same. And that was our intention. We aimed to study the effect of the addition of the whole extract of the herbs, but not the specific compounds isolated from them. We realised that the effect on the basis of specific compounds could be completely different from the use of the whole herbal extract. In order to make the text of the manuscript completely clear in this regard, we have made some additions to the introduction, the methodology, and the discussion of the results.

Round 2

Reviewer 2 Report

The author could improve the DOE and resubmit the article based on the reviewer's specific request to have a specific/targeted compound that could affect the food-based material. 

Author Response

Author's Reply to the Review Report (Reviewer 2)

Reviewer 2: The author could improve the DOE and resubmit the article based on the reviewer's specific request to have a specific/targeted compound that could affect the food-based material. 

Answer: Thank you for your comments. We have corrected the DOE in the manuscript and indicate the target groups of compounds that may have affected the food samples tested.

In this topic area we have added comments to the following chapters: "1. Introduction", "2.1. Materials", "2.2. Agar well diffusion method of evaluating antimicrobial activity of herbal extracts ", "2.3.1. Fermentation of cream and fermentation curves of cream ", "3. Results and discussion", "3.12. Oxidative stability", and "4. Conclusions".

We hope that the current version of the manuscript, especially the DOE description, is satisfactory.
